# WHO IS YOUR RIGHT MIXUP PARTNER IN POSITIVE AND UNLABELED LEARNING

**Changchun Li**[1,*], **Ximing Li**[1,*,†], **Lei Feng**[2,3], **Jihong Ouyang**[1,*]
[1]College of Computer Science and Technology, Jilin University, China
[2]College of Computer Science, Chongqing University, China
[3]Imperfect Information Learning Team, RIKEN Center for Advanced Intelligence Project, Japan
{changchunli93,liximing86}@gmail.com,lfeng@cqu.edu.cn,ouyj@jlu.edu.cn

## ABSTRACT

Positive and Unlabeled (PU) learning targets inducing a binary classifier from weak training datasets of positive and unlabeled instances, which arise in many real-world applications. In this paper, we propose a novel PU learning method, namely Positive and unlabeled learning with Partially Positive Mixup ($P^3$Mix), which simultaneously benefits from data augmentation and supervision correction with a heuristic mixup technique. To be specific, we take inspiration from the decision boundary deviation phenomenon observed in our preliminary experiments, where the learned PU boundary tends to deviate from the fully supervised boundary towards the positive side. For the unlabeled instances with ambiguous predictive results, we select their mixup partners from the positive instances around the learned PU boundary, so as to transform them into augmented instances near to the boundary yet with more precise supervision. Accordingly, those augmented instances may push the learned PU boundary towards the fully supervised boundary, thereby improving the classification performance. Comprehensive experimental results demonstrate the effectiveness of the heuristic mixup technique in PU learning and show that $P^3$Mix can consistently outperform the state-of-the-art PU learning methods.

## 1 INTRODUCTION

**P**ositive and **U**nlabeled (**PU**) learning refers to a specific binary classification problem, where only a small number of positive training instances are manually annotated but all other instances are unlabeled (Liu et al., 2002). Such kind of datasets naturally arise in many significant real-world scenarios such as product recommendation (Hsieh et al., 2015), deceptive reviews detection (Ren et al., 2014), and medical diagnosis (Yang et al., 2012). For specific example, many diseases, *e.g.,* Alzheimer's disease, Amyotrophic Lateral Sclerosis, and Parkinson's disease, are very infrequent and with long latency, hence only few diagnosed patients are known but a much larger population of undiagnosed individuals may be either diseased or healthy. Treating the diagnosed ones as positive instances and the undiagnosed ones as unlabeled instances results in such PU datasets of medical diagnosis. To meet those practical demands, PU learning has drawn increasing interest from the machine learning community (Bekker & Davis, 2020).

Formally, let $\mathbf{x} \in \mathbb{R}^d$ and $y \in \{0, 1\}$ be the feature representation and category label, respectively, where the positive instance is indicated by $y = 1$ and the negative one by $y = 0$. In the context of PU learning, the training dataset is composed of the sets of positive instances $\mathcal{P} = \{(\mathbf{x}_i, y_i = 1)\}_{i=1}^{n_p}$ and unlabeled instances $\mathcal{U} = \{\mathbf{x}_i\}_{i=n_p+1}^{n_p+n_u}$, where $\mathcal{U}$ contains both positive and negative instances. The target is to learn a binary classifier based on such weak training dataset $\mathcal{P} \cup \mathcal{U}$.

During the past decades, many PU learning methods have been proposed, where, naturally, the essential idea is to estimate the negative instances from the set of unlabeled instances $\mathcal{U}$. Generally, most of existing PU learning methods can be divided into two categories, termed as sample-selection

---

*Key Laboratory of Symbolic Computation and Knowledge Engineering of Ministry of Education, China
†Corresponding author.

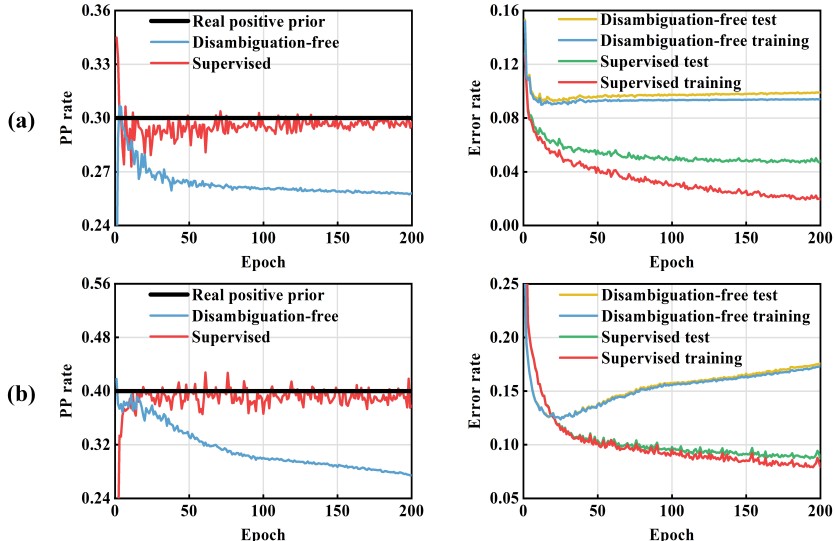

Figure 1: Rates of training instances Predicted as Positive (PP rate) and error rates of disambiguation-free[1] and fully supervised objectives on (a) FashionMNIST and (b) CIFAR-10 (Xiao et al., 2017; Krizhevsky, 2016; Chen et al., 2020a). The disambiguation-free objective suffers from much lower PP rates than the real positive prior as well as the fully supervised objective. This implies the decision boundary deviation phenomenon, which results in higher error rates.

methods and cost-sensitive methods. The sample-selection methods, as the name suggests, mainly select reliable negative instances from $\mathcal{U}$ using various heuristic strategies, *e.g.,* Naïve Bayes (Liu et al., 2002), $k$NN (Zhang & Zuo, 2009), $k$-means (Chaudhari & Shevade, 2012), and reinforcement learning (Luo et al., 2021); and then apply supervised methods over positive and those reliable negative instances. In contrast, the cost-sensitive methods treat all unlabeled instances as corrupted negative ones, and correct the estimation bias of the objective by employing well-designed misclassification risks such as unbiased risk estimator (du Plessis et al., 2014; 2015; Kiryo et al., 2017) and maximum margin loss (Shi et al., 2018; Gong et al., 2019b; Zhang et al., 2019; Gong et al., 2019a).

Orthogonal to the aforementioned techniques, we note that some PU learning methods such as (Chen et al., 2020a; Wei et al., 2020) have made preliminary attempts to integrate with the art of mixup, *i.e.,* an economic-yet-effective data augmentation method (Zhang et al., 2018). Formally, mixup generates an augmented instance $(\widehat{\mathbf{x}}, \widehat{y})$ with the convex combination of any pair of instances $\{(\mathbf{x}_i, y_i), (\mathbf{x}_j, y_j)\}$ drawn from the training dataset:

$$\widehat{\mathbf{x}} = \lambda\mathbf{x}_i + (1 - \lambda)\mathbf{x}_j, \quad \widehat{y} = \lambda y_i + (1 - \lambda)y_j, \qquad \lambda \sim \text{Beta}(\alpha, \alpha), \ \alpha \in (0, \infty).$$

Previous studies have indicated that mixup is approximately equivalent to applying adversarial training (Zhang et al., 2021), enabling to improve robustness with even scarce and noisy supervision (Thulasidasan et al., 2019; Carratino et al., 2020; Zhang et al., 2021). Accordingly, it has been successfully used to solve various learning problems with weak supervision, *e.g.,* semi-supervised learning (Berthelot et al., 2019), noisy label learning (Li et al., 2020b), and partial label learning (Yan & Guo, 2020).

**Our story and contribution.** Inspired by the recent success of mixup in learning with weak supervision, our original goal is to thoroughly investigate the impact of mixup in PU learning. To this end, we begin with a naive *disambiguation-free* objective of PU learning, where all unlabeled instances are treated as *pseudo-negative instances*, denoted by $\widetilde{\mathcal{U}} = \{(\mathbf{x}_i, y_i = 0)\}_{i=n_p+1}^{n_p+n_u}$, and the binary classifier is trained based on $\mathcal{P} \cup \widetilde{\mathcal{U}}$. In preliminary experiments, we found an interesting phenomenon, where the number of training instances predicted as positive by the disambiguation-free classifier tends to be smaller than usual as illustrated in Fig.1. This phenomenon implies that

---

[1]Specially, we apply the early learning regularization (Liu et al., 2020) into the objective to keep it stable.

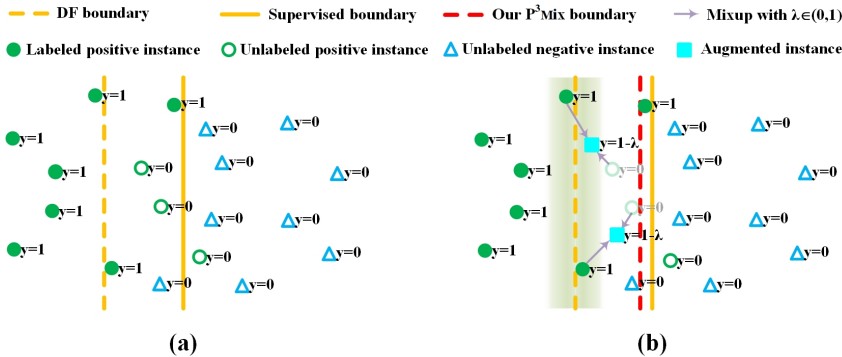

Figure 2: Toy examples of (a) the decision boundary deviation phenomenon and (b) the proposed heuristic mixup for marginal pseudo-negative instances. Best viewed in color.

the disambiguation-free boundary tends to deviate from the fully supervised boundary towards the positive side, expressed by a toy example shown in Fig.2(a). We consider that the decision boundary deviation is mainly caused by the *marginal pseudo-negative instances*, which lie between the two boundaries. Such instances are more likely to be positive but actually annotated by negative. Motivated by this observation, we extend mixup to a specific heuristic version for PU learning, enabling to achieve data augmentation and supervision correction simultaneously. Its basic idea is to transform the marginal pseudo-negative instances into augmented instances which are partially positive and yet also lie between the two boundaries, so as to push the learned boundary towards the fully supervised one. This can be achieved by selecting the mixup partners for marginal pseudo-negative instances from the positive instances that are around the learned boundary, as expressed in Fig.2(b). With this insight, we propose a novel PU method, namely **P**ositive and unlabeled learning with **P**artially **P**ositive **Mix**up (**P$^3$Mix**). Generally, P$^3$Mix is easy-to-implement, where, specifically, we can define the marginal pseudo-negative instances using the predictive results and the positive instances around the boundary using the entropy values of predictive results. To evaluate the effectiveness of P$^3$Mix, we conduct a number of experiments on benchmark datasets. Experimental results demonstrate that P$^3$Mix can consistently outperform the state-of-the-art PU methods.

## 2 THE PROPOSED P$^3$MIX METHOD

In this section, we introduce the proposed **P$^3$Mix** method for PU learning. We first revisit and clarify some important notations: the set of positive instances $\mathcal{P} = \{(\mathbf{x}_i, y_i = 1)\}_{i=1}^{n_p}$ and the set of unlabeled instances $\mathcal{U} = \{\mathbf{x}_i\}_{i=n_p+1}^{n_p+n_u}$. By treating all unlabeled instances as negative, we translate $\mathcal{U}$ into the set of pseudo-negative instances $\widetilde{\mathcal{U}} = \{(\mathbf{x}_i, y_i = 0)\}_{i=n_p+1}^{n_p+n_u}$. Given batches $\mathcal{X}_p \subset \mathcal{P}$ and $\mathcal{X}_u \subset \widetilde{\mathcal{U}}$, the disambiguation-free objective of PU learning can be formulated as follows:

$$\mathcal{L}(\mathcal{X}_p, \mathcal{X}_u; \boldsymbol{\Theta}) = \frac{1}{|\mathcal{X}_p|} \sum_{(\mathbf{x},y)\in\mathcal{X}_p} \ell\big(f(\mathbf{x};\boldsymbol{\Theta}), y\big) + \frac{\beta}{|\mathcal{X}_u|} \sum_{(\mathbf{x},y)\in\mathcal{X}_u} \ell\big(f(\mathbf{x};\boldsymbol{\Theta}), y\big), \qquad (1)$$

where $f(\cdot;\boldsymbol{\Theta})$ is a trainable neural network, *i.e.,* the binary classifier, parameterized by $\boldsymbol{\Theta}$; $\ell(\cdot,\cdot)$ is the loss function; and $\beta$ is the coefficient parameter.

To achieve data augmentation and supervision correction simultaneously, P$^3$Mix transforms $\mathcal{X}_p$ and $\mathcal{X}_u$ into the batches of augmented instances $\widehat{\mathcal{X}}_p$ and $\widehat{\mathcal{X}}_u$ using the proposed heuristic mixup technique. Accordingly, the objective of P$^3$Mix is then expressed as follows:

$$\mathcal{L}(\widehat{\mathcal{X}}_p, \widehat{\mathcal{X}}_u; \boldsymbol{\Theta}) = \frac{1}{|\widehat{\mathcal{X}}_p|} \sum_{(\widehat{\mathbf{x}},\widehat{y})\in\widehat{\mathcal{X}}_p} \ell\big(f(\widehat{\mathbf{x}};\boldsymbol{\Theta}), \widehat{y}\big) + \frac{\beta}{|\widehat{\mathcal{X}}_u|} \sum_{(\widehat{\mathbf{x}},\widehat{y})\in\widehat{\mathcal{X}}_u} \ell\big(f(\widehat{\mathbf{x}};\boldsymbol{\Theta}), \widehat{y}\big), \qquad (2)$$

$$\widehat{\mathcal{X}}_p, \widehat{\mathcal{X}}_u = \text{HeuristicMixup}(\mathcal{X}_p, \mathcal{X}_u, \alpha), \qquad (3)$$

where $\alpha \in (0, \infty)$ is a hyperparameter of mixup. Next, we describe the details of heuristic mixup.

---

**Algorithm 1** Training procedure of $P^3$Mix, $P^3$Mix-E and $P^3$Mix-C

---

**Input:**

$\mathcal{P} \cup \mathcal{U}$: training instances; $\beta$: coefficient parameter; $\gamma$: thresholding parameter; $k$: size of the candidate mixup pool; $\alpha$: hyperparameter of mixup; $\eta$: coefficient parameter of early-learning regularization                                                                    $\triangleright \eta = 0$ for $P^3$Mix and $P^3$Mix-C

**Output:**

$\Theta$: binary classifier parameters

1: **Initialize** $\Theta$, the mean-teacher parameters $\widetilde{\Theta}$ and the candidate mixup pool $\mathcal{X}_{cnd}$ randomly, translate $\mathcal{U}$ into $\widetilde{\mathcal{U}}$;
2: **for** $t = 1, 2, \cdots, MaxEpoch$ **do**
3:     Shuffle $\mathcal{P} \cup \widetilde{\mathcal{U}}$ into $I$ mini-batches and denote the $i$-th mini-batch by $(\mathcal{X}_p^i, \mathcal{X}_u^i)$;
4:     **for** $i = 1, 2, \cdots, I$ **do**
5:         Estimate marginal pseudo-negative instances $\mathcal{X}_{mpn}$ using Eq.(6);
6:         Select the mixup partners for each instance within $\mathcal{X}_p^i \cup \mathcal{X}_u^i$ using Eq.(5);
7:         Set labels of $\{(\mathbf{x}, y = 0) | (\mathbf{x}, y = 0) \in \mathcal{X}_u^i, f(\mathbf{x}; \Theta) > \gamma\}$ to 1; $\triangleright$ Optional for $P^3$Mix-C
8:         Construct $\widehat{\mathcal{X}}_p^i \cup \widehat{\mathcal{X}}_u^i$ by applying Eq.(4) to $\mathcal{X}_p^i \cup \mathcal{X}_u^i$ and their mixup partners;
9:         Estimate $\{\widetilde{\mathbf{y}}_j\}_{j=1}^{|\widehat{\mathcal{X}}_p^i| + |\widehat{\mathcal{X}}_u^i|}$ for instances in $\widehat{\mathcal{X}}_p^i \cup \widehat{\mathcal{X}}_u^i$ by $f(\cdot; \widetilde{\Theta})$;     $\triangleright$ Optional for $P^3$Mix-E
10:         Update $\Theta$ by $\nabla_{\Theta}\big(\mathcal{L}(\widehat{\mathcal{X}}_p^i, \widehat{\mathcal{X}}_u^i; \Theta) + \eta \mathcal{R}_{elr}(\{(\widehat{\mathbf{x}}_j, \widetilde{\mathbf{y}}_j)\}_{j=1}^{|\widehat{\mathcal{X}}_p^i| + |\widehat{\mathcal{X}}_u^i|}; \Theta)\big)$ with Adam;
11:     **end for**
12:     Update $\mathcal{X}_{cnd}$ using Eq.(7);
13:     Update $\widetilde{\Theta}$ by $\Theta$ with the move-average;                                                $\triangleright$ Optional for $P^3$Mix-E
14: **end for**

---

## 2.1 TRAINING WITH HEURISTIC MIXUP

Basically, for each instance $(\mathbf{x}_i, y_i) \in \mathcal{X}_p \cup \mathcal{X}_u$ we select a mixup partner $(\mathbf{x}_j, y_j)$ to generate an augmented instance $(\widehat{\mathbf{x}}_i, \widehat{y}_i)$ using the modified mixup operator[2](Berthelot et al., 2019):

$$\widehat{\mathbf{x}}_i = \lambda' \mathbf{x}_i + (1 - \lambda')\mathbf{x}_j, \quad \widehat{y}_i = \lambda' y_i + (1 - \lambda')y_j, \quad \lambda' = \max(\lambda, 1 - \lambda),$$
$$\lambda \sim \text{Beta}(\alpha, \alpha), \quad \alpha \in (0, \infty), \quad (4)$$

accordingly forming the augmented instance sets $\widehat{\mathcal{X}}_p$ and $\widehat{\mathcal{X}}_u$.

Our heuristic mixup refers to a guidance of mixup partner selection to refine the imprecise supervision within $\mathcal{X}_u$. We take inspiration from the phenomenon, where the boundary learned by Eq.(1) tends to deviate from the fully supervised boundary towards the positive side as illustrated in Fig.2(a). The marginal pseudo-negative instances $\mathcal{X}_{mpn} \subset \mathcal{X}_u$ lie between the two boundaries, and they are more likely to be positive but actually annotated by negative. To resolve this problem, for each of them we uniformly select a mixup partner from the *candidate mixup pool* $\mathcal{X}_{cnd} \subset \mathcal{P}$ of positive instances that are around the current learned boundary, so as to generate an augmented instance which is partially positive and yet also lies between the two boundaries as expressed in Fig.2(b). Besides, for positive instances $\mathcal{X}_p$ and other pseudo-negative instances $\mathcal{X}_u \setminus \mathcal{X}_{mpn}$, we uniformly choose their mixup partners from $\mathcal{X}_p \cup \mathcal{X}_u$. The overall mixup partner selection is formulated as follows:

$$(\mathbf{x}_j, y_j) \sim \begin{cases} \text{Uniform}(\mathcal{X}_{cnd}) & \text{if } (\mathbf{x}_i, y_i) \in \mathcal{X}_{mpn}, \\ \\ \text{Uniform}(\mathcal{X}_p \cup \mathcal{X}_u) & \text{if } (\mathbf{x}_i, y_i) \in \mathcal{X}_p \cup \mathcal{X}_u \setminus \mathcal{X}_{mpn}. \end{cases} \quad (5)$$

In what follows, we introduce how to estimate the marginal pseudo-negative instances $\mathcal{X}_{mpn}$ and construct the candidate mixup pool $\mathcal{X}_{cnd}$.

---

[2]Because we compute individual loss terms for positive instances and pseudo-negative ones in Eq.(2) appropriately, we define $\lambda' = \max(\lambda, 1 - \lambda)$ to guarantee that the feature of each augmented instance $\widehat{\mathbf{x}}_i$ is closer to $\mathbf{x}_i$ than the mixup partner $\mathbf{x}_j$. Consequently, $(\widehat{\mathbf{x}}_i, \widehat{y}_i)$ is assigned into $\widehat{\mathcal{X}}_p$ if $(\mathbf{x}_i, y_i) \in \mathcal{X}_p$, or $\widehat{\mathcal{X}}_u$ otherwise.

**Marginal pseudo-negative instance estimation.** Because the fully supervised boundary is exactly unknown, we have to estimate the set of marginal pseudo-negative instances $\mathcal{X}_{mpn}$ from $\mathcal{X}_u$. In this work, we define them as the "unreliable" pseudo-negative instances measured by the predictive scores with thresholding parameter $\gamma \in [0.5, 1]$:

$$\mathcal{X}_{mpn} = \big\{(\mathbf{x}, y = 0) | (\mathbf{x}, y = 0) \in \mathcal{X}_u, 1 - \gamma \leq f(\mathbf{x}; \mathbf{\Theta}) \leq \gamma \big\}, \tag{6}$$

where $\gamma = 0.5$ implies $\mathcal{X}_{mpn} = \emptyset$, and $\gamma = 1$ means $\mathcal{X}_{mpn} = \mathcal{X}_u$.

**Candidate mixup pool.** We maintain a candidate mixup pool $\mathcal{X}_{cnd}$ containing the positive instances around the current learned boundary from $\mathcal{P}$. To be specific, for each positive instance we compute its entropy value of the predictive score, and update the candidate mixup pool with the top-$k$ positive instances as follows:

$$\mathcal{X}_{cnd} = \big\{(\mathbf{x}, y = 1) | (\mathbf{x}, y = 1) \in \mathcal{P}, \mathcal{H}(f(\mathbf{x}; \mathbf{\Theta})) \in \text{Rank}(\{\mathcal{H}(f(\mathbf{x}_i; \mathbf{\Theta}))\}_{i=1}^{n_p}) \big\}, \tag{7}$$

where $\mathcal{H}(\cdot)$ is the entropy, and $\text{Rank}(\cdot)$ outputs a set of positive instances with the top-$k$ maximum entropy values. For efficiency, we update $\mathcal{X}_{cnd}$ per-epoch. The full training procedure is shown in *Algorithm 1*.

## 2.2 ROBUSTNESS

The augmented instances within $\widehat{\mathcal{X}}_u$ also suffer from imprecise supervision even using the heuristic mixup. To make P³Mix more robust, we employ two tricks, *i.e.,* early-learning regularization (Liu et al., 2020) and pseudo-negative instance correction. We call the versions with early-learning regularization and pseudo-negative instance correction as **P³Mix-E** and **P³Mix-C**, respectively.

**Early-learning regularization.** We employ the early learning regularization to prevent the memorization of imprecise supervision (Liu et al., 2020). For each mixup instance within $\widehat{\mathcal{X}}_p \cup \widehat{\mathcal{X}}_u$, we estimate an auxiliary target vector $\widetilde{\mathbf{y}}$, and formulate the early-learning regularization below:

$$\mathcal{R}_{elr}(\{(\widehat{\mathbf{x}}_i, \widetilde{\mathbf{y}}_i)\}_{i=1}^{|\widehat{\mathcal{X}}_p| + |\widehat{\mathcal{X}}_u|}; \mathbf{\Theta}) = \frac{1}{|\widehat{\mathcal{X}}_p| + |\widehat{\mathcal{X}}_u|} \sum_{i=1}^{|\widehat{\mathcal{X}}_p| + |\widehat{\mathcal{X}}_u|} \log\big(1 - \langle f(\widehat{\mathbf{x}}_i; \mathbf{\Theta}), \widetilde{\mathbf{y}}_i \rangle\big) \tag{8}$$

Here, we estimate the target vector $\widetilde{\mathbf{y}}$ of each instance by using the mean teacher technique (Tarvainen & Valpola, 2017), and incorporate Eq.(8) to Eq.(2).

**Pseudo-negative instance correction.** We concentrate on the pseudo-negative instances with high confidence to be positive $\big\{(\mathbf{x}, y = 0) | (\mathbf{x}, y = 0) \in \mathcal{X}_u, f(\mathbf{x}; \mathbf{\Theta}) > \gamma\big\}$. We directly revise their labels to positive before their corresponding mixup operators.

## 3 EXPERIMENT

### 3.1 EXPERIMENTAL SETTINGS

**Datasets.** In the experiments, we employ three prevalent benchmark datasets, including Fashion-MNIST (F-MNIST) (Xiao et al., 2017),[3] CIFAR-10 (Krizhevsky, 2016),[4] and STL-10 (Coates et al., 2011).[5] The dataset statistics are described in Table 1. Note that all benchmark datasets have 10 category labels, and we denote them with integers ranging from 0 to 9 following the default settings in torchvision 0.10.0. For each dataset, we group those category labels into two disjoint sets as positive or negative, and generate two synthetic PU datasets by reversing the definitions of positive and negative labels. Following the protocol of (Chen et al., 2020a), the specific definitions of labels ("positive" vs "negative") are as follows: F-MNIST-1: "1,4,7" vs "0,2,3,5,6,8,9", F-MNIST-2: "0,2,3,5,6,8,9" vs "1,4,7"; CIFAR-10-1: "0,1,8,9" vs "2,3,4,5,6,7", CIFAR-10-2: "2,3,4,5,6,7" vs "0,1,8,9"; STL-10-1: "0,2,3,8,9" vs "1,4,5,6,7", STL-10-2: "1,4,5,6,7" vs "0,2,3,8,9". For each dataset, we randomly select 1,000 positive instances from the training set, and 500 instances as the validation set.

---

[3]https://github.com/zalandoresearch/fashion-mnist
[4]http://www.cs.toronto.edu/~kriz/cifar.html
[5]https://cs.stanford.edu/~acoates/stl10

Table 1: Specification of datasets and corresponding backbones.

| Dataset | #Train | #Test | Input size | Backbone |
|---------|--------|-------|------------|----------|
| F-MNIST | 60,000 | 10,000 | 28×28 | LeNet-5 |
| CIFAR-10 | 50,000 | 10,000 | 3×32×32 | 7-layer CNN |
| STL-10 | 105,000 | 8,000 | 3×96×96 | 7-layer CNN |

Table 2: Results of classification accuracy (mean±std). The highest scores among PU learning methods are indicated in **bold**.

| Dataset | F-MNIST-1 | F-MNIST-2 | CIFAR-10-1 | CIFAR-10-2 | STL-10-1 | STL-10-2 |
|---------|-----------|-----------|------------|------------|----------|----------|
| uPU | 71.3±1.4 | 84.0±4.0 | 76.5±2.5 | 71.6±1.4 | 76.7±3.8 | 78.2±4.1 |
| nnPU | 89.7±0.8 | 88.8±0.9 | 84.7±2.4 | 83.7±0.6 | 77.1±4.5 | 80.4±2.7 |
| nnPU+mixup | 91.4±0.3 | 88.2±0.7 | 87.2±0.6 | 85.8±1.2 | 79.8±0.8 | 82.2±0.9 |
| Self-PU | 90.8±0.4 | 89.1±0.7 | 85.1±0.8 | 83.9±2.6 | 78.5±1.1 | 80.8±2.1 |
| PAN | 88.7±1.2 | 83.6±2.5 | 87.0±0.3 | 82.8±1.0 | 77.7±2.5 | 79.8±1.4 |
| VPU | 90.6±1.2 | 86.8±0.8 | 86.8±1.2 | 82.5±1.1 | 78.4±1.1 | 82.9±0.7 |
| MIXPUL | 87.5±1.5 | 89.0±0.5 | 87.0±1.9 | 87.0±1.1 | 77.8±0.7 | 78.9±1.9 |
| PULNS | 90.7±0.5 | 87.9±0.5 | 87.2±0.6 | 83.7±2.9 | 80.2±0.8 | 83.6±0.7 |
| P³Mix-E | **91.9±0.3** | **89.5±0.5** | 88.2±0.4 | 84.7±0.5 | 80.2±0.9 | **83.7±0.7** |
| P³Mix-C | **92.0±0.4** | **89.4±0.3** | **88.7±0.4** | **87.9±0.5** | **80.7±0.7** | **84.1±0.3** |
| Supervised | 95.2±0.2 | 95.2±0.2 | 91.3±0.3 | 91.3±0.3 | 85.6±0.6 | 85.6±0.6 |

**Baseline methods.** To verify the effectiveness of $P^3$Mix, we utilize eight PU learning baselines, including uPU (du Plessis et al., 2014), nnPU (Kiryo et al., 2017), nnPU+mixup, Self-PU (Chen et al., 2020b), PAN (Hu et al., 2021), VPU (Chen et al., 2020a), MIXPUL (Wei et al., 2020) and PULNS (Luo et al., 2021), as well as the supervised method for comparison. The corresponding implementation details of baselines are present in Appendix A. For all comparing methods, we adopt the classifiers (including the discriminator of PAN) by LeNet-5 for F-MNIST, and 7-layer CNN for CIFAR-10 and STL-10. Specially, the baseline methods of uPU, nnPU and Self-PU require the prior knowledge of class proportion, however, the prior is actually unknown for STL-10 since it contains many "real" unlabeled instances. Accordingly, we estimate the class proportion of STL-10 by using the SOTA KM2 method (Ramaswamy et al., 2016) before evaluating uPU, nnPU, and Self-PU.

**Implementation details of $P^3$Mix.** We implement $P^3$Mix, $P^3$Mix-E and $P^3$Mix-C by using Pytorch (Paszke et al., 2019) with the Adam algorithm (Kingma & Ba, 2014). We employ the cross entropy function as the loss function $\ell$ of Eq.(2), fix the mixup hyperparameter $\alpha$ to 1 and the size $k$ of the candidate mixup pool $\mathcal{X}_{cnd}$ to 100, and choose the coefficient parameter $\beta$ from $\{0.8, 0.9, 1.0\}$, the thresholding parameter $\gamma$ from $\{0.85, 0.9, 0.95\}$. We will make the sensitivity analysis on $\{\beta, \gamma\}$ later. Specially, the early-learning regularization parameter of $P^3$Mix-E is chosen from $\{1.0, 2.0, 3.0, 4.0, 5.0\}$.

## 3.2 CLASSIFICATION PERFORMANCE

For each dataset, we independently run each comparing method 5 times and report the average classification accuracy in Table 2. Generally, our $P^3$Mix-E and $P^3$Mix-C consistently outperform all PU learning baselines on all benchmark datasets, indicating their superior performance. Compared with nnPU+mixup, VPU and MIXPUL, which utilize the typical mixup technique, both $P^3$Mix-E and $P^3$Mix-C achieve significant performance gain in most cases, *i.e.,* about $1\% \sim 5\%$ improvements. These results imply that our proposed heuristic mixup benefits to the supervision correction within marginal pseudo-negative instances. Compared with the three cost-sensitive PU learning methods uPU, nnPU and Self-PU, $P^3$Mix-E and $P^3$Mix-C also outperform them by about $1\% \sim 4\%$ in all cases. Besides, we observe that all discriminative PU learning methods except uPU perform better than the GAN-based PU learning method PAN in most cases. The possible reason is that the imprecise supervision within unlabeled instances makes the identification of fake instances for the discriminator more difficult, resulting in a worse classifier.

Table 3: Results of ablative study (mean±std). The highest scores are indicated in **bold**.

| Dataset | F-MNIST-1 | F-MNIST-2 | CIFAR-10-1 | CIFAR-10-2 | STL-10-1 | STL-10-2 |
|---|---|---|---|---|---|---|
| DF | 75.2±1.2 | 62.7±2.8 | 72.0±3.2 | 57.4±3.7 | 78.1±0.6 | 80.6±2.4 |
| DF+mixup | 78.4±1.7 | 72.4±1.4 | 79.2±3.0 | 67.4±2.5 | 78.9±0.3 | 80.7±1.9 |
| $P^3$Mix | 87.0±1.1 | 79.0±1.6 | 87.0±1.1 | 84.3±0.6 | 79.8±0.7 | 83.4±0.7 |
| DF-E | 90.1±0.7 | 74.2±5.5 | 82.4±1.6 | 69.4±3.0 | 67.3±2.0 | 75.0±3.7 |
| DF-E+mixup | 90.6±0.7 | 86.1±2.5 | 85.7±0.7 | 76.4±0.9 | 78.3±1.1 | 79.3±2.3 |
| $P^3$Mix-E | **91.9±0.3** | **89.5±0.5** | **88.2±0.4** | **84.7±0.5** | **80.2±0.9** | **83.7±0.7** |
| DF-C | 89.6±1.8 | 87.4±2.4 | 87.2±0.8 | 84.7±1.1 | 80.2±3.0 | 82.7±2.6 |
| DF-C+mixup | 91.6±0.3 | 88.3±1.2 | 87.7±1.1 | 81.3±3.6 | 79.9±3.1 | 81.6±2.8 |
| $P^3$Mix-C | **92.0±0.4** | **89.4±0.3** | **88.7±0.4** | **87.9±0.5** | **80.7±0.7** | **84.1±0.3** |

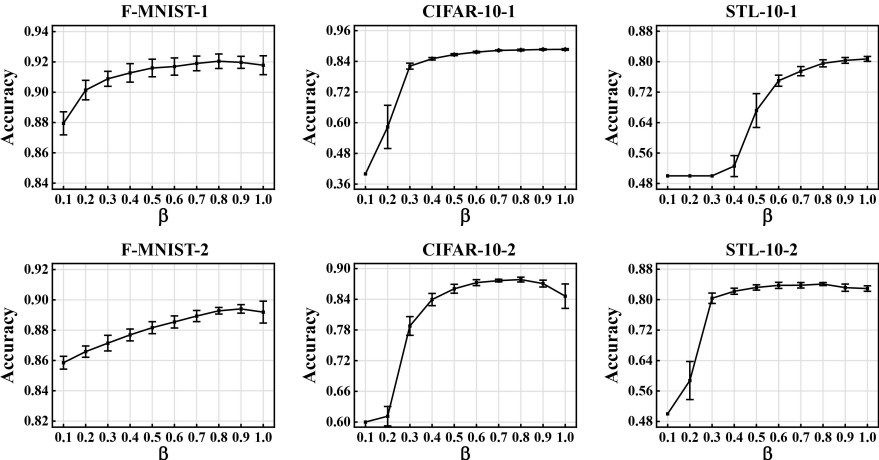

Figure 3: Sensitivity analysis of the coefficient parameter $\beta$.

## 3.3 ABLATION STUDY

To evaluate the effectiveness of our proposed heuristic mixup, we conduct the ablation experiments on all benchmark datasets. Specifically, we compare $P^3$Mix, $P^3$Mix-E and $P^3$Mix-C with the Disambiguation-Free (DF) objective of Eq.(1), DF+mixup, augmented by the typical mixup technique, and their versions with early-learning regularization ("-E") and pseudo-negative instance correction ("-C"). The experimental results are reported in Table 3. It clearly demonstrates that the proposed heuristic mixup can significantly improve the classification performance. This result is expected because the heuristic mixup can simultaneously achieve data augmentation and supervision correction by refining the imprecise supervision within marginal pseudo-negative instances. Besides, we can also observe that both early-learning regularization and pseudo-negative instance correction contribute to the improvement of the classification performance in all cases, proving the effectiveness of those two tricks in improving the robustness of models.

## 3.4 SENSITIVITY ANALYSIS

In this section, we examine the sensitivities of the coefficient parameter $\beta$ and the thresholding parameter $\gamma$.

**Sensitivity of $\beta$.** We examine the impact of different $\beta$ values over the set $\{0.1, 0.2, \cdots, 1.0\}$ by $P^3$Mix-C and plot the experimental results in Fig.3. We omit the results of $P^3$Mix and $P^3$Mix-E due to their similar performance curves and also page-limitation. Obviously, the performance achieves the highest and is relatively stable when $\beta \geq 0.8$, and it sharply drops as the values become smaller especially on CIFAR-10-1, CIFAR-10-2, STL-10-1 and STL-10-2. Notice that the coefficient parameter $\beta$ is used to balance the importance of the positive and pseudo-negative parts in Eq.(2). The larger or smaller value of $\beta$ will result in the indecent importance of the pseudo-negative part, leading to a unstable classifier. Therefore, we suggest tuning $\beta$ over the set $\{0.8, 0.9, 1.0\}$ in practice.

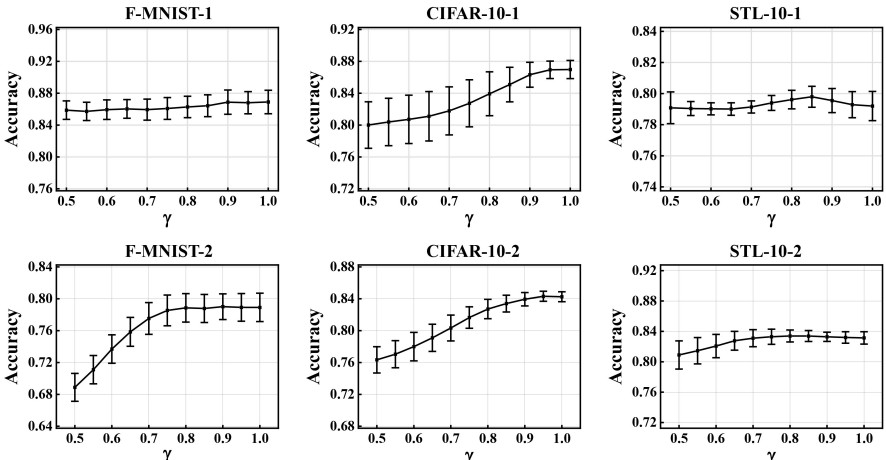

Figure 4: Sensitivity analysis of the thresholding parameter $\gamma$.

**Sensitivity of $\gamma$.** Generally, the thresholding parameter $\gamma$ is utilized to estimate the set of marginal pseudo-negative instances $\mathcal{X}_{mpn}$ from pseudo-negative instances $\mathcal{X}_u$. However, both the early-learning regularization and pseudo-negative instance correction may be affected by $\gamma$ since they are mainly used to control the corrupted negative instances within $\mathcal{X}_u \setminus \mathcal{X}_{mpn}$. Accordingly, we perform the sensitivity analysis of $\gamma$ by P³Mix. Specifically, we vary the value of $\gamma$ over the set $\{0.5, 0.55, \cdots, 1.0\}$. As shown in Fig.4, P³Mix achieves the best performance when $\gamma \in \{0.85, 0.9, 0.95\}$, and performs relatively worse when $\gamma$ is too small or too large. This result is expected because the smaller value of $\gamma$ implies that $\mathcal{X}_{mpn}$ contains fewer marginal pseudo-negative instances, *e.g.*, $\mathcal{X}_{mpn} = \emptyset$ when $\gamma = 0.5$, and it will hurt the supervision correction of heuristic mixup. Besides, when $\gamma$ becomes too large, too many pseudo-negative instances will be treated as marginal pseudo-negative ones, *e.g.*, $\mathcal{X}_{mpn} = \mathcal{X}_u$ when $\gamma = 1.0$, resulting in the reduction of the variety of augmented instances. In summary, its suggested setting is given by $\{0.85, 0.9, 0.95\}$.

## 4 RELATED WORK

In this section, we review the representative studies on PU learning, especially the ones most related to P³Mix. Besides, we briefly introduce the recent studies of mixup for data augmentation.

### 4.1 PU LEARNING

Weakly supervised learning (Zhou, 2018) mainly tackles datasets with weak supervision, such as incomplete labels (van Engelen & Hoos, 2020; Li et al., 2021a), inexact labels (Feng et al., 2020a;b; Li & Wang, 2020; Li et al., 2020a; 2021b), and inaccurate labels (Li et al., 2020b; Nguyen et al., 2020). PU learning is an emerging paradigm of weakly supervised learning. The early PU learning works focus on the sample-selection paradigm. As the name suggests, the basic idea of sample-selection methods is to select reliable negative instances from unlabeled instances to form pseudo-binary dataset before applying supervised methods. Existing methods have proposed various heuristic strategies for negative sample selection, *e.g.*, 1-DNF (Yu et al., 2002; 2004; Peng et al., 2008), Naïve Bayes (Liu et al., 2002), Rocchio extraction (Li & Liu, 2003), $k$NN (Zhang & Zuo, 2009), $k$-means (Chaudhari & Shevade, 2012), large margin method (Gong et al., 2018) and reinforcement learning (Luo et al., 2021). Early works based on sample-selection spirit mainly focus on exploiting various traditional classification and clustering approaches to construct the heuristic strategy. Recently, PULNS (Luo et al., 2021) employs a negative selector trained by a reinforcement learning framework, where the selector, the selection of negative instances, and the performance of the classifier are treated as the agent, action and reward, respectively. Then the classifier is induced from the mixture of positive instances and negative ones selected by the selector.

In contrast, the community of PU learning has recently paid more attention to the cost-sensitive methods, which directly treat all unlabeled instances as corrupted negative instances and correct

the estimation bias of the objective by employing well-designed misclassification risks. The uPU (du Plessis et al., 2014; 2015) made an early attempt of unbiased risk estimation, which reformulates the misclassification risk as an equivalent and ubiased form depending only on PU datasets. However, as reported in (Kiryo et al., 2017), the risk of uPU would become negative due to overfitting when using the flexible and complex models such as deep networks. To remedy this issue, the authors of (Kiryo et al., 2017) suggest the nnPU method, *i.e.,* the non-negative version of uPU. Besides, the Self-PU (Chen et al., 2020b) further considers the learning capability of the model itself, and jointly employs three self-supervision techniques, *i.e.,* a self-paced strategy to discover confident positive and negative instances, a self-calibrated instance-aware loss to explore meaningful supervision over unconfident instances, and a self-supervision consistency by teacher-students learning. Other cost-sensitive methods are based on the maximum margin objective, and refine the bias of corrupted negative instances by various tricks, such as unbiased centroid estimation of unlabeled instances (Shi et al., 2018; Gong et al., 2019b), label calibration with a hat loss (Gong et al., 2019a), and margin-based label disambiguation (Zhang et al., 2019).

In parallel with the aforementioned methods, several GAN-based PU learning methods (Hou et al., 2018; Chiaroni et al., 2018; Guo et al., 2020; Na et al., 2020; Hu et al., 2021) have been proposed. The GenPU (Hou et al., 2018) generates positive and negative instances, and induces a classifier from those generated instances. The PAN (Hu et al., 2021) is based on adversarial learning on the probability distributions of a discriminator and a classifier.

Orthogonal to those PU learning methods, our $P^3$Mix concentrates on data augmentation and extends the well-established mixup technique (Zhang et al., 2018) to a specific heuristic version for PU learning, enabling to achieve data augmentation and supervision correction simultaneously. The recent PU learning works most related to $P^3$Mix are VPU (Chen et al., 2020a) and MixPUL (Wei et al., 2020), in which the mixup is used as a regularization to improve the robustness of classifiers and performed among labeled and unlabeled instances randomly. In contrast, our $P^3$Mix constructs a heuristic mixup partner selection to refine the imprecise supervision within unlabeled instances.

### 4.2 MIXUP AUGMENTATION

The mixup technique (Zhang et al., 2018) generates augmented instances with convex combinations of training instances. Despite its simplicity, it can effectively improve the robustness with even scarce and noisy supervision (Thulasidasan et al., 2019; Carratino et al., 2020; Zhang et al., 2021). Further, a number of modified mixup versions (Verma et al., 2019; Yun et al., 2019; Guo et al., 2019; Kim et al., 2020; Hendrycks et al., 2020) have been proposed, *e.g.,* manifold mixup that generates convex combinations on the latent feature space (Verma et al., 2019) and puzzle mixup that further explores the saliency information and underlying statistics of instances (Kim et al., 2020). In this work, we extend the typical mixup to a specific heuristic version for PU learning.

## 5 CONCLUSION

In this paper, we propose a novel PU learning method named $P^3$Mix, which extends the typical mixup technique to a heuristic version. The story begins with the observation of the decision boundary deviation phenomenon, which inspires us to propose a guidance of mixup partners selection, especially for the marginal pseudo-negative instances. Fortunately, this heuristic mixup technique can simultaneously achieve data augmentation and supervision correction for PU learning. Generally, our $P^3$Mix is easy-to-implement, and we also employ two tricks to improve the robustness of $P^3$Mix. We compare $P^3$Mix against a number of existing PU learning methods on benchmark datasets. Experimental results show the superior performance of $P^3$Mix and the effectiveness of the heuristic mixup technique.

### ACKNOWLEDGMENTS

We would like to acknowledge support for this project from the National Key R&D Program of China (No.2021ZD0112501, No.2021ZD0112502), the National Natural Science Foundation of China (NSFC) (No.61876071, No.62006094), the Key R&D Projects of Science and Technology Department of Jilin Province of China (No.20180201003SF, No.20190701031GH).

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

## A  DETAILS OF BASELINE METHODS

Eight existing PU learning baselines and the supervised method are employed for comparison in this paper. The details of baseline methods are presented below.

- **u**nbiased **PU** learning (**uPU**) (du Plessis et al., 2014): A cost-sensitive method based on unbiased risk estimation. We use the public code from the net.[6]

- **n**on-**n**egative **PU** learning (**nnPU**) (Kiryo et al., 2017): A cost-sensitive method based on non-negative risk estimation. We use the public code from the net.[6] [suggested settings: $\beta = 0$ and $\gamma = 1.0$]

- **nnPU+mixup**: A cost-sensitive method incorporating the typical mixup technique into the nnPU method by mixing positive instances and unlabeled ones separately.

- **Self-PU** (Chen et al., 2020b): A cost-sensitive method with self-supervision scheme. We use the public code from the net.[7] [suggest settings: $\alpha = 10.0$, $\beta = 0.3$, $\gamma = 1/16$, Pace1 $= 0.2$ and Pace2 $= 0.3$]

- **P**redictive **A**dversarial **N**etworks (**PAN**) (Hu et al., 2021): A GAN-based PU learning method with a discriminator and a classifier. We use the public code from the net.[8] [suggested settings: $\lambda = 1e - 4$]

- **V**ariational **PU** learning (**VPU**) (Chen et al., 2020a): A PU learning method based on the variational principle. We use the public code from the net.[9] [suggested settings: $\alpha = 0.3$, $\beta \in \{1e - 4, 3e - 4, 1e - 3, \cdots, 1, 3\}$]

- **MIXPUL** (Wei et al., 2020): A PU learning method based on the consistency regularization with the mixup technique. We use the public code from the net.[10] [suggested settings: $\alpha = 1.0$, $\beta = 1.0$, $\eta = 1.0$]

- **P**ositive-**U**nlabeled **L**earning with effective **N**egative sample **S**elector (**PULNS**) (Luo et al., 2021): A sample-selection method with reinforcement learning. We implement an in-house python code with a 3-layer MLP selector suggested by the paper. [suggested settings: $\alpha = 1.0$ and $\beta \in \{0.4, 0.6, 0.8, 1.0\}$]

- **Supervised**: The classifiers trained on the fully supervised datasets.

---

[6]https://github.com/kiryor/nnPUlearning
[7]https://github.com/TAMU-VITA/Self-PU
[8]https://github.com/morning-dews/PAN
[9]https://github.com/HC-Feynman/vpu
[10]https://github.com/Stomach-ache/MixPUL

Table 4: Results of classification accuracy (mean±std) on the credit card fraud detection dataset. The highest scores among PU learning methods are indicated in **bold**.

| Metric | Accuracy | Precision | Recall |
|--------|----------|-----------|--------|
| uPU | 97.0±0.2 | 96.5±3.6 | 83.4±1.3 |
| nnPU | 98.4±0.1 | 97.4±1.1 | 83.4±1.3 |
| nnPU+mixup | 98.1±0.1 | 96.0±3.2 | 82.9±1.6 |
| Self-PU | **99.2±0.1** | 92.4±3.4 | 85.8±2.0 |
| PAN | 99.1±0.1 | 98.5±1.0 | 85.4±1.3 |
| VPU | 98.6±0.5 | **99.7±0.6** | 84.9±5.7 |
| MixPUL | 98.4±0.3 | 79.2±3.5 | 86.6±1.3 |
| PULNS | 99.0±0.1 | 95.6±1.9 | 83.2±2.1 |
| $\text{P}^3\text{Mix-E}$ | 99.0±0.1 | 96.5±1.8 | **87.7±2.0** |
| $\text{P}^3\text{Mix-C}$ | 98.8±0.1 | 94.1±1.2 | 86.5±1.8 |

## B    ADDITIONAL EXPERIMENTAL RESULTS ON REALWORLD DATASET

To examine the performance of our proposed $\text{P}^3\text{Mix-E}$ and $\text{P}^3\text{Mix-C}$ in practice, we perform the experiments on the Credit Card Fraud Detection task of Kaggle[11]. We utilize a subset of the original Credit Card Fraud Detection dataset, which contains all (492) fraudulent instances and 10000 genuine ones selected randomly from the original dataset. In this subset, the proportion of the positive instances (frauds) is about 0.0469. We use 20% of the constructed subset as the test dataset, and all others as the training one of PU learning, in which 100 frauds are selected randomly as positive instances. The accuracy, precision and recall are utilized as metrics. Table 4 reports the average results of independently running 5 times of all comparison methods. Overall, our proposed $\text{P}^3\text{Mix-E}$ and $\text{P}^3\text{Mix-C}$ gain the best recall score, and also achieve the competitive performance on the accuracy and precision scores, even the dataset is highly unbalanced.

---

[11]https://www.kaggle.com/mlg-ulb/creditcardfraud

