# OpenReview forum: "Who Is Your Right Mixup Partner in Positive and Unlabeled Learning"
_ICLR.cc/2022/Conference — ICLR 2022 Poster_

### Official Review · Reviewer_eqge · 2021-10-27

**Correctness:** 3
**Technical Novelty And Significance:** 3
**Empirical Novelty And Significance:** 3
**Recommendation:** 8
**Confidence:** 4

**Main Review:**

Strengths:
1. The paper is well written and easy-to-follow.
2. The motivation and the proposed method are clearly described. Especially, the observed phenomenon and the key idea of correcting marginal pseudo-negative examples with a heuristic mixup technique is interesting, and may potentially be useful for semi-supervised learning.
3. The proposed PU learning approach does not require explicit computation of a class prior.
4. The experiments are well-conducted, and comprehensively compared to recent SOTA methods. Ablations and sensitivity analysis are also shown.

Weakness:
1. Does the proposed approach rely on the "selected completely at random" (SCAR) assumption?
2. Showing sensitivity analysis on \alpha would be better.
3. Why is the size of the candidate mixup pool fixed as 100? Bigger candidate mixup pool, better performance?
4. Could the authors release the code?


**Summary Of The Paper:**

In this paper, the authors focus on the problem of positive and unlabeled learning. They show an interesting phenomenon, where the learned PU boundary tends to deviate the supervised boundary towards the positive side when treating unlabeled examples as pseudo-negative examples. The phenomenon may imply there are a number of marginal pseudo-negative examples that are more likely to be positive but labeled as negative. Based on this, the paper proposes a PU learning approach building on a novel heuristic mixup technique, which can achieve both data augmentation and supervision correction. They also present many empirical results to show the superior performance comparing with SOTA PU learning methods.

**Summary Of The Review:**

The problem is interesting and the proposed method is promising. I vote for accepting this paper.

---

> ### Author Response · Authors · 2021-11-16
> **Response to Reviewer eqge**
>
> Thank you for your insightful comments.
>
> We have revised and updated the manuscript. Our replies are listed below.
>
>
>
> Q1: Does the proposed approach rely on the "selected completely at random" (SCAR) assumption?
>
> A1: Our proposed approach does not rely on the SCAR assumption. In this paper, we just consider correcting the supervision of unlabeled instances with the proposed heuristic mixup, which does not rely on the data distribution assumption.
>
>
>
> Q2: Showing sensitivity analysis on \alpha would be better.
>
> A2: Thank you for your suggestion. In the paper, we focus on the selection of the mixup partners for the training instances to achieve data augmentation and supervision correction simultaneously. Thus $\alpha$ is not the major hyper-parameter of our proposed approach. Otherwise, we agree with your opinion. As you suggested, we also perform the experiments on F-MNIST-1 by varying the values of $\alpha$. The experimental results are shown in the following table.
>
> | $\alpha$   | 0.2          | 1.0          | 2.0          | 4.0          | 8.0          | 16.0         |
> | ---------- | ------------ | ------------ | ------------ | ------------ | ------------ | ------------ |
> | P$^3$MIX-C | 91.6$\pm$0.6 | 92.0$\pm$0.4 | 91.8$\pm$0.4 | 91.2$\pm$0.3 | 90.6$\pm$0.4 | 90.4$\pm$0.5 |
>
> From the above table, P$^3$MIX-C performs better and is relatively stable when $\alpha\leq2.0$. Note that $\alpha$ is the parameter of Beta distribution in the mixup, smaller $\alpha$ is more likely to generate the mixing coefficient $\lambda$ around 0.1, and in this paper we hope that for the marginal pseudo-negative instances their augmented instances are partially positive and yet also lie between the two boundaries as expressed in Fig.2(b) of the paper, in other words, the features of their augmented instances are still closer to ones of the marginal pseudo-negative instances. Besides, we wish that the heuristic mixup can also produce the “newer” instances (different with observed instances) with other instances (reliable instances). Thus, we fix $\alpha=1.0$, and the choice is consistent with the above experimental results. And its suggesting setting is given by [1.0, 2.0].
>
>
>
> Q3: Why is the size of the candidate mixup pool fixed as 100? Bigger candidate mixup pool, better performance?
>
> A3: We argue that the bigger candidate mixup pool may hurt the classification performance. As mentioned in A2, in this paper we hope that for the marginal pseudo-negative instances their augmented instances are partially positive and yet also lie between the two boundaries as expressed in Fig.2(b) of the paper, in other words, the features of their augmented instances are still closer to ones of the marginal pseudo-negative instances. Thus the labeled positive instances of the candidate mixup pool should be closer to the marginal pseudo-negative instances, i.e, around the current learned boundary as expressed in Fig.2(b) of the paper. In the paper, we select the positive instances for the candidate mixup pool by ranking the entropy values of their predictive scores. To avoid the candidate mixup pool containing too many positive instances far away from the marginal pseudo-negative instances, we fix the size of the candidate mixup pool as 100 following our early experiments. The bigger candidate mixup pool would contain too many positive instances far away from the marginal pseudo-negative instances, which may hurt the supervision correction for the marginal pseudo-negative instances, resulting in worse classification performance.
>
>
>
> Q4: Could the authors release the code?
>
> A4: We have updated our code in the supplementary material.

---

### Official Review · Reviewer_XfpN · 2021-11-01

**Correctness:** 3
**Technical Novelty And Significance:** 2
**Empirical Novelty And Significance:** 3
**Recommendation:** 6
**Confidence:** 4

**Main Review:**

The ideas of marginal pseudo negative instances and candidate mixup pool are interesting. Since PU learning is different from ordinary supervised learning, it would be reasonable to develop a specific mixup approach for PU learning.

---

It is not really clear to me that when to use the proposed early-learning regularization. If possible, it is nice to write when to use early-learning regularization (and also, pseudo-negative instance correction) in Algorithm 1 or create Algorithm 2 including the techniques in Section 2.2 (Robustness).

It is expected that the proposed method would be compared with one of the PU learning methods, e.g., nnPU, with the ordinary mixup technique. Probably, we can respectively augment P and U data by a mixup technique and then use the existing PU learning methods. This approach can be regarded as a simple baseline against the proposed method. It is expected that such a simple baseline would be included in the experiments.

In Section 3.4, it is reported that \beta \in {0.8, 0.85, 0.9, 0.95} is better on the basis of the experiments. But, it is not clear the relation between the class-prior probability and \beta. When we set \beta=1/(2\pi) and the sigmoid loss is used, minimization of Eq. (1) is equivalent to minimization of the objective function of uPU. It is also known that a rough class-prior estimation is sufficient when the true-class prior is known to be large (du Plessis et al., NeurIPS2014). The suggested \beta might not be generalized to the settings other than the settings of this paper. That is, the suggest \beta might correspond to the rough estimation of the class-prior, by chance. To support the effectiveness of the suggested \beta, it would be necessary to show the comprehensive experiments to illustrate the relation between \beta and the class-prior.

In Section 2.1, it is not obvious the rule of assigning the generated samples into \hat{X}\_p or \hat{X}\_u. Since this part is very important in the proposed method, without explicitly-written rules, it would be difficult to reproduce the results from this paper.

In Figure 1, it seems that "disambiguation-free" is labeled as "discrimination-free."

**Summary Of The Paper:**

This paper proposes a variant of the mixup technique for positive-unlabeled learning. Based on the observation that the learned PU boundary tends to deviate towards the positive side, the authors suggest selecting samples between the learned PU and supervised boundaries. The proposed P3MIX method and its variant improve the classification performance of PU learning.

**Summary Of The Review:**

This paper proposes a mixup method specialized for PU learning. The idea of marginal pseudo-negative instance estimation is interesting. In the current manuscript, there are however several unclear points and it lacks a simple baseline in the experiments. Having such a simple baseline, the advantage of the proposed method will become more clear.

---

> ### Author Response · Authors · 2021-11-15
> **Response to Reviewer XfpN**
>
> Thank you for your constructive comments.
>
> We revised our manuscript based on your advice and continue to reflect your advice on our manuscript. Our replies are listed below.
>
> Q1: It is not really clear to me that when to use the proposed early-learning regularization. If possible, it is nice to write when to use early-learning regularization (and also, pseudo-negative instance correction) in Algorithm 1 or create Algorithm 2 including the techniques in Section 2.2 (Robustness).
>
> A1: We agree with your point. We revised Algorithm 1 to include the robustness techniques of Section 2.2 in the newest manuscript.
>
>
>
> Q2: It is expected that the proposed method would be compared with one of the PU learning methods, e.g., nnPU, with the ordinary mixup technique.
>
> A2: We agree with your opinion. As you suggested, we added nnPU+mixup as a baseline in our newest manuscript. The corresponding results can be found in Table 2 in the newest manuscript and the following table.
>
> | Dataset      | F-MNIST-1      | F-MNIST-2      | CIFAR-10-1     | CIFAR-10-2     | STL-10-1       | STL-10-2       |
> | ------------ | -------------- | -------------- | -------------- | -------------- | -------------- | -------------- |
> | nnPU+mixup   | 91.4$\pm$0.3 | 88.2$\pm$0.7 | 87.2$\pm$0.6 | 85.8$\pm$1.2 | 79.8$\pm$0.8 | 82.2$\pm$0.9 |
> | P$^3$MIX-E | 91.9$\pm$0.3 | 89.5$\pm$0.5 | 88.2$\pm$0.4 | 84.7$\pm$0.5 | 80.2$\pm$0.9 | 83.7$\pm$0.7 |
> | P$^3$MIX-C | 92.0$\pm$0.4 | 89.4$\pm$0.3 | 88.7$\pm$0.4 | 87.7$\pm$0.5 | 80.7$\pm$0.7 | 84.1$\pm$0.3 |
>
> As shown in the above table, our proposed P3MIX-E and P3MIX-C perform better than nnPU+mixup in most cases, especially P3MIX-C, which consistently outperforms nnPU+mixup by about 1%--2% on all datasets. These results demonstrate again that our proposed heuristic mixup benefits to the supervision correction within marginal pseudo-negative instances. Otherwise, we have performed the ablation study to show the effectiveness of our heuristic mixup in Section 3.3 of the paper, in which similar results can be observed.
>
>
>
> Q3: It is not clear the relation between the class-prior probability and \beta. To support the effectiveness of the suggested \beta, it would be necessary to show the comprehensive experiments to illustrate the relation between \beta and the class-prior.
>
> A3: We clarify that the coefficient $\beta$ doesn’t have a relation with the class-prior probability, and in this paper it is merely utilized to balance the positive and pseudo-negative parts in the objective function. We clarify that the most interesting point of our paper is the proposed heuristic mixup technique inspired by the decision boundary deviation phenomenon observed in our preliminary experiments. And we will explore the application of the proposed heuristic mixup on other PU methods, such uPU and nnPU, and other weak supervised learning tasks. Otherwise, we agree with your opinion, thus perform a experiment on F-MNIST-2 by varying the values of $\beta$, due to the time-limitation the experimental results on CIFAR-10-2 and STL-10-2 will be added in the future manuscript.  The accuracy results of F-MNIST-2 are shown in the following table.
>
> |       $\beta$        | 0.1  | 0.2  | 0.3  | 0.4  | 0.5  | 0.6  | 0.7  | 0.8  | 0.9  | 1.0  |
> | :--------------------: | :--: | :--: | :--: | :--: | :--: | :--: | :--: | :--: | :--: | :--: |
> | F-MNIST-1($\pi$=0.3) | 87.5 | 90.0 | 90.8 | 91.3 | 91.4 | 91.5 | 91.8 | 91.7 | 91.8 | 92.0 |
> | F-MNIST-2($\pi$=0.7) | 85.8 | 86.6 | 87.1 | 87.7 | 88.2 | 88.6 | 88.9 | 89.2 | 89.4 | 89.4 |
>
> From the results in the above table, no matter big or small the class-prior probability is, the classification performance becomes better when $\beta$ is bigger. Besides, as shown in the sensitive experiments on $\beta$ in Section 3.4 of the paper, the performance achieves the highest and is relatively stable when $\beta$ ≥ 0.8. These results are not consistent with $\beta=1/(2\pi)$.
>
>
>
> Q4: In Section 2.1, it is not obvious the rule of assigning the generated samples into $\hat{\mathcal{X}}_p$ or $\hat{\mathcal{X}}_u$.
>
> A4: Thank you for your suggestion. As you suggested, we clarified the rule of assigning the augmented instances into $\widehat{\mathcal{X}}_p$ or  $\widehat{\mathcal{X}}_u$ in the newest manuscript. In this paper, we utilize the motified mixup technique with $\lambda'=\max(\lambda,1-\lambda)$ to guarantee that the feature of each augmented instance $\mathbf{\widehat{x}}_i$ is closer to $\mathbf{x}_i$ than the mixup partner $\mathbf{x}_j$. Thus, if $(\mathbf{x}_i,y_i)$ is a labeled positive instance, its augmented instance $(\mathbf{\widehat{x}}_i,\widehat{y}_i)$ is assigned into $\widehat{\mathcal{X}}_p$, or $\widehat{\mathcal{X}}_u$ otherwise.

---

> > ### Comment · Reviewer_XfpN · 2021-11-17
> > **Questions have been resolved**
> >
> > Thank you for your responses.
> > I read the answers and the revised manuscript. Now, Q1-4 have been resolved.
> > I was concerned that the suggested $\beta$ would not work well when $\pi$ is small. Although the results to answer Q3 do not show the performance of the other methods and the performance on the other datasets due to time limitations, it seems that the suggested $\beta$ will work well even when small/large class-prior probabilities. I thus increased my score. To support the effectiveness of the suggested $\beta$ more, I expect the authors to add extensive results to show the comparison of the proposed method with the existing methods over various $\beta$ and $\pi$ into the final version.

---

> > > ### Author Response · Authors · 2021-11-17
> > > **Response to Reviewer XfpN for Questions have been resolved**
> > >
> > > Thank you! We will add the corresponding comparison experiments in the further manuscript.

---

### Official Review · Reviewer_x3p8 · 2021-11-01

**Correctness:** 3
**Technical Novelty And Significance:** 3
**Empirical Novelty And Significance:** 3
**Recommendation:** 8
**Confidence:** 3

**Main Review:**

The idea that achieves data augmentation and supervision correction by refining mixup is interesting and easy to implement. This paper is well-organized and well-motivated. Extensive experiments on several datasets and ablation studies prove the effectiveness of the proposed method and its components. Exhaustive discussion of related work is presented.

Concerns:
1.	The method is evaluated on several image classification datasets. It will be better to do experiments on practical PU learning problems, such as product recommendation and medical diagnosis, as mentioned in the paper. The data distributions of these practical problems may be different from image dataset. Thus, the conclusions may be different.
2.	MixPUL is a recently proposed PU learning method, which also applies mixup. It may be better to compare with MixPUL and discuss the difference.
3.	What is the main difference between positive and unlabeled (PU) learning and anomaly detection?
4.	Does the proposed method also work well with Cutmix[1] augmentation?

[1] Cutmix: Regularization strategy to train strong classifiers with localizable features, Yun et al 2019


**Summary Of The Paper:**

This paper studies an interesting weakly supervised binary classification problem called positive and unlabeled (PU) learning. The authors propose a novel PU learning method inspired by the boundary deviation phenomenon observed in experiments. Specifically, a new mixup method is proposed, which selects the mixup partners for unlabeled examples heuristically to obtain more correct supervised signals. Extensive empirical results, including ablation study and sensitiveness analysis, are provided to evaluate the proposal.

**Summary Of The Review:**

The paper proposes a new positive-unlabeled learning method which achieves data augmentation and supervision correction simultaneously by refining mixup strategy. The method is well-motivated by empirical findings. Extensive experiments and ablation study demonstrate the effectiveness of the proposed method.

---

> ### Author Response · Authors · 2021-11-16
> **Response to Reviewer x3p8 (1-2)**
>
> Q3: What is the main difference between positive and unlabeled (PU) learning and anomaly detection?
>
> A3: To the best of our knowledge, the main differences between PU learning and anomaly detection include: (1) From the data clustering assumption[1]: PU learning concentrates on the classification, the positive (or negative) instances are similar to each other, thus a concept could be found for them; in contrast, anomaly detection focuses on filtering the instances (named anomalies) that are significantly dissimilar to the normal ones, or located further away from the bulk of data points, besides the anomalies are always diversified, so they can rarely group into one cluster.
>
> (2) From the task types[2]: There are three categories of anomaly detection, including unsupervised anomaly detection, supervised anomaly detection, and semi-supervised anomaly detection. PU learning is only somewhat similar to semi-supervised anomaly detection. And many methods based on PU learning are proposed for semi-supervised anomaly detection, such as [3], [4].
>
> Q4: Does the proposed method also work well with Cutmix augmentation?
>
> A4: We merely concentrate on choosing the right mixup partner for each instance in PU learning, so how to mixup does not care in this paper, and the proposed method could also work well with Cutmix augmentation.
>
>
>
> [1] Zhang, Ya-Lin, et al. Anomaly detection with partially observed anomalies. Companion Proceedings of The Web Conference. 2018.
>
> [2] Chandola, V.; Banerjee, A.; Kumar, V. (2009). Anomaly detection: A survey. ACM Computing Surveys. 41 (3): 1–58.
>
> [3] Zhang, Jiaqi, et al. Positive and unlabeled learning for anomaly detection with multi-features. Proceedings of the 25th ACM international conference on Multimedia. 2017.
>
> [4] Li, Xiao-Li, et al. Positive unlabeled learning for data stream classification. Proceedings of the 2009 SIAM international conference on data mining. Society for Industrial and Applied Mathematics, 2009.

---

> > ### Comment · Reviewer_x3p8 · 2021-11-18
> > **Questions have been clearly answered.**
> >
> > Thank you for the response! My four questions have been clearly answered. Especially, experiment results have been provided about the practical PU dataset and MixPUL, which further enhance the paper.

---

> > > ### Author Response · Authors · 2021-11-19
> > > **Response to Questions have been clearly answered.**
> > >
> > > Thanks! We are happy the response is helpful.

---

> ### Author Response · Authors · 2021-11-16
> **Response to Reviewer x3p8 (1-1)**
>
> Thank you for your insightful comments. The experiments of comparison with MixPUL you suggested are very helpful, and we have included them in the newest manuscript. Due to the time limitation, a part of the experimental results on real-world PU datasets are shown here,  and the full results will be added to the further manuscript.  Our replies are listed below.
>
> Q1: The method is evaluated on several image classification datasets. It will be better to do experiments on practical PU learning problems, such as product recommendation and medical diagnosis, as mentioned in the paper. The data distributions of these practical problems may be different from image dataset. Thus, the conclusions may be different.
>
> A1: We agree with your opinion. As you suggested, we perform the experiments on Credit Card Fraud Detection dataset from Kaggle. We utilize a subset of the original Credit Card Fraud Detection dataset, which contains all (492) fraudulent instances and 10000 genuine instances selected randomly from the original dataset. In this subset, the proportion of the positive instances (frauds) is about 0.0469. We use 20% of the constructed subset as the test dataset and remain as the training one. We randomly select 100 positive instances from the training dataset.  Due to the time limitation, we only perform the experiments of uPU, nnPU, DF-C, DF-C+mixup, P$^3$MIX-C, and show the results in the following table.
>
> | Method     | Accuracy     | Precision    | Recall       |
> | ---------- | ------------ | ------------ | ------------ |
> | uPU        | 97.0$\pm$0.2 | 96.5$\pm$3.6 | 38.0$\pm$3.8 |
> | nnPU       | 98.4$\pm$0.1 | 97.4$\pm$1.1 | 67.4$\pm$2.6 |
> | DF-C       | 96.2$\pm$1.5 | 59.3$\pm$5.7 | 79.6$\pm$4.1 |
> | DF-C+mixup | 98.1$\pm$0.5 | 93.6$\pm$1.1 | 64.5$\pm$5.6 |
> | P$^3$MIX-C | 98.8$\pm$0.1 | 91.6$\pm$1.2 | 80.6$\pm$1.2 |
>
> As shown in the above table, overall, our proposed P$^3$MIX-C performs best on the metrics of accuracy and recall. Note that in the subset used in the experiment, the positive instances (frauds) account for about 0.0469 of all instances. That means that the training dataset are highly unbalanced, thus the accuracy of all comparison methods are very close. Otherwise, though the precision scores of uPU and nnPU are higher than one of P$^3$MIX-C, the credit card fraud detection task aims to find as many as possible frauds, and the recall is a more important metric. Our P$^3$MIX-C beats uPU and nnPU by 42.6% and 13.2%, respectively, on the recall score. Besides, one can also observe that our P$^3$MIX-C outperforms DF-C and DF-C+mixup, only has a small decrease compared with DF-C+mixup on precision. These results demonstrate again the effectiveness of the proposed heuristic mixup.
>
>
>
> Q2: MixPUL is a recently proposed PU learning method, which also applies mixup. It may be better to compare with MixPUL and discuss the difference.
>
> A2: Thank you for your suggestion. As you suggested, we added the experiments of comparison with MixPUL in the newest manuscript, and also show the experimental results in the following table.
>
> | Dataset    | F-MNIST-1    | F-MNIST-2    | CIFAR-10-1   | CIFAR-10-2   | STL-10-1     | STL-10-2     |
> | ---------- | ------------ | ------------ | ------------ | ------------ | ------------ | ------------ |
> | MixPUL     | 87.5$\pm$1.5 | 89.0$\pm$0.5 | 87.0$\pm$1.9 | 87.0$\pm$1.1 | 77.8$\pm$0.7 | 78.9$\pm$1.9 |
> | P$^3$MIX-E | 91.9$\pm$0.3 | 89.5$\pm$0.5 | 88.2$\pm$0.4 | 84.7$\pm$0.5 | 80.2$\pm$0.9 | 83.7$\pm$0.7 |
> | P$^3$MIX-C | 92.0$\pm$0.4 | 89.4$\pm$0.3 | 88.7$\pm$0.4 | 87.7$\pm$0.5 | 80.7$\pm$0.7 | 84.1$\pm$0.3 |
>
> As shown in the above table, we can observe that our P$^3$MIX-E and P$^3$MIX-C outperform MixPUL in most cases. For example, on STL-10-1 and STL-10-2, our P$^3$MIX-E and P$^3$MIX-C beat MixPUL by about 3%–5%. Note that MixPUL merely employs the typical mixup technique for consistency regularization, in contrast, P$^3$MIX-E and P$^3$MIX-C employ the proposed heuristic mixup to achieve data augmentation and supervision correction simultaneously. These results show the superiority of our proposed heuristic mixup over the typical mixup technique.

---

### Official Review · Reviewer_fDKn · 2021-11-02

**Correctness:** 3
**Technical Novelty And Significance:** 3
**Empirical Novelty And Significance:** 3
**Recommendation:** 6
**Confidence:** 4

**Details Of Ethics Concerns:**

No ethics concerns.

**Main Review:**

Strengths
+ The motivation of this paper is strong, and the research problem is interesting. The authors found a phenomenon that the decision boundary learned with PU data tends to shift to the positive side compared to the boundary learned with PN data. Because shifting of the decision boundary leads to the bias of learned classifiers, the authors try to reduce the bias by exploiting a heuristic mixup technique.
+ The proposed method has strong empirical performance. The experimental results on different datasets show that the proposed method can consistently outperform the state-of-the-art PU learning methods.
+ This paper is well structured and easy to follow.

Weaknesses
+ It seems that “directional boundary” is not commonly used in existing papers. To avoid confusion, it is better to add a specific definition or change to other words. I assume that it is the same as the “decision boundary”.
+ To select the marginal pseudo-negative instance, predictive scores can be different by employing different learning models. It is better to add some discussion that what should pay attention to when choosing the learning model for estimating predictive scores.
+ In the abstract, “For the unlabelled instances with ambiguous predictive results…”. The word “ambiguous” is not clear. I think that the authors should high-levelly explain the word “ambiguous”.

**Summary Of The Paper:**

The authors proposed to reduce the bias of classifiers learned on PU data by a heuristic mixup technique that partially selects the unlabelled instances and mixes them up with the positive instances around the decision boundary learned with PU data. The experimental results demonstrate the effectiveness of the heuristic mixup technique.

**Summary Of The Review:**

This paper has a strong motivation as I mentioned above. I like the idea of using a heuristic mixup technique for reducing bias in PU learning. I think the idea that adds heuristic to mixup could be extended to other weakly supervised machine learning tasks.

---

> ### Author Response · Authors · 2021-11-15
> **Response to Reviewer fDKn**
>
> Thank you for your insightful comments.
>
> We have revised and updated the manuscript. Our replies are listed below.
>
>
>
> Q1: To select the marginal pseudo-negative instance, predictive scores can be different by employing different learning models. It is better to add some discussion that what should pay attention to when choosing the learning model for estimating predictive scores.
>
> A1: We agree with your point. We think that the quality of the features extracted by the learning model should be paid attention to when choosing the learning model for estimating predictive scores. More discriminative features could make the prediction to be easy, resulting in more accurate predictive scores, as well as more accurate marginal pseudo-negative instances.  Otherwise, we argue that the threshold $\gamma$ for selecting the marginal pseudo-negative instances with predictive scores is also important. Though the predictive scores could be different by employing different learning models, the instances, whose predictive scores are higher than a threshold (or lower than another threshold), can be treated as reliable positive (or negative) instances, and the remaining “unreliable” instances can be chosen as the marginal pseudo-negative ones. Thus, the threshold decides how many the real marginal pseudo-negative instances are selected, and how many the fake ones are included. Here, we hope that the selected marginal pseudo-negative instances contain more real marginal pseudo-negative instances, and less fake ones (a few fake ones merely be augmented by mixed with labeled positive instances, and will have little effect), because the marginal pseudo-negative instances just are mixed with a few labeled positive instances, excessive fake marginal pseudo-negative instances will ruin the variety of augmented instances. In the paper, we have performed the sensitive analysis of the threshold $\gamma$.
>
>
>
> Q2: In the abstract, “For the unlabelled instances with ambiguous predictive results…”. The word “ambiguous” is not clear. I think that the authors should high-levelly explain the word “ambiguous”.
>
> A2: Thank you for your suggestion. We explain the word “ambiguous” in the following. Here, the unlabeled instances with ambiguous predictive results mean the marginal pseudo-negative instances, which are more likely to be positive but actually annotated by negative. Following the decision boundary deviation phenomenon observed in the paper, where the learned PU boundary tends to deviate from the fully supervised boundary towards the positive side, the predictive scores of these instances may be near 0.5, such as 0.48, then it is positive with 0.48 probability and negative with 0.52 probability. Thus, one can’t decide that if these instances actually are positive or negative from their predictive scores. So we say their predictive results are ambiguous.

---

> > ### Comment · Reviewer_fDKn · 2021-11-30
> > **Final score**
> >
> > Thanks for your responses.
> >
> > My questions have been clearly answered. I think this paper is above the acceptance threshold, and I would like to keep my score.
> >
> > Best Regards,
> >
> > Reviewer fDKn

---

### Decision · Program_Chairs · 2022-01-20

**Decision:**

Accept (Poster)

**Comment:**

Mixup is very helpful when the training sample is scarce or has weak supervision. The paper studies how to adapt mixup to positive and unlabeled (PU) learning, a representative weakly supervised learning problem. By studying the specific properties of PU learning, the authors propose the concept of marginal pseudo-negative instances, which are more likely to be positive but actually annotated by negative. A novel mixup variant has been proposed for PU learning by mixuping the marginal pseudo-negative instances with the positive instance around the classification boundary. The effectiveness has been empirically shown.